# The Anti-Calcitonin Gene-Related Peptide (Anti-CGRP) Antibody Fremanezumab Reduces Trigeminal Neurons Immunoreactive to CGRP and CGRP Receptor Components in Rats

**DOI:** 10.3390/ijms241713471

**Published:** 2023-08-30

**Authors:** Birgit Vogler, Annette Kuhn, Kimberly D. Mackenzie, Jennifer Stratton, Mária Dux, Karl Messlinger

**Affiliations:** 1Institute of Physiology and Pathophysiology, Friedrich-Alexander-University, D-91054 Erlangen, Germany; birgit.vogler@fau.de (B.V.); annette.kuhn@fau.de (A.K.); 2Teva Pharmaceuticals, Redwood City, CA 94063, USA; kimberly.mackenzie01@gmail.com (K.D.M.); jennifer_stratton@mac.com (J.S.); 3Department of Physiology, University of Szeged, H-6720 Szeged, Hungary; dux.maria@med.u-szeged.hu

**Keywords:** fremanezumab, monoclonal antibody, calcitonin gene-related peptide, trigeminal ganglion, CGRP release, rat, migraine pain

## Abstract

Treatment with the anti-CGRP antibody fremanezumab is successful in the prevention of chronic and frequent episodic migraine. In preclinical rat experiments, fremanezumab has been shown to reduce calcitonin gene-related peptide (CGRP) release from trigeminal tissues and aversive behaviour to noxious facial stimuli, which are characteristic pathophysiological changes accompanying severe primary headaches. To further decipher the effects of fremanezumab that underlie these antinociceptive effects in rats, immunohistochemistry and ELISA techniques were used to analyse the content and concentration of CGRP in the trigeminal ganglion, as well as the ratio of trigeminal ganglion neurons which are immunoreactive to CGRP and CGRP receptor components, 1–10 days after subcutaneous injection of fremanezumab (30 mg/kg) compared to an isotype control antibody. After fremanezumab treatment, the fraction of trigeminal ganglion neurons which were immunoreactive to CGRP and the CGRP receptor components calcitonin receptor-like receptor (CLR) and receptor activity modifying protein 1 (RAMP1) was significantly lowered compared to the control. The content and concentration of CGRP in trigeminal ganglia were not significantly changed. A long-lasting reduction in CGRP receptors expressed in trigeminal afferents may contribute to the attenuation of CGRP signalling and antinociceptive effects of monoclonal anti-CGRP antibodies in rats.

## 1. Introduction

The monoclonal anti-CGRP antibody fremanezumab is one of three monoclonal antibodies targeting CGRP that are successfully used in the prophylaxis of chronic and frequent episodic migraine [1,2,3]. The antibodies target calcitonin gene-related peptide (CGRP), a potent vasodilatory neuropeptide, which is released during migraine attacks and trigemino-autonomic headaches from trigeminal afferents [4]. CGRP release is not only symptomatic for these types of primary headaches, but CGRP can also induce similar headache states when it is infused into patients suffering from these headaches [5,6]. The pathophysiological mechanisms underlying the nociceptive effect of CGRP have not been fully elucidated, although based on animal models, it has been hypothesized that CGRP has a cross-activating effect on primary trigeminal afferents [7,8,9]. In short, the basic idea of these hypotheses is that CGRP released from primary afferents in the meninges or the trigeminal ganglion is activating another type of (not CGRP-releasing) afferents, directly or via glial cells that produce excitatory substances like nitric oxide [10,11,12]. Even though these hypotheses may theoretically explain an acute therapeutic action through blocking of CGRP signalling, the long-lasting antinociceptive effect after one single application of monoclonal anti-CGRP antibodies requires additional exploration. The slow elimination of these antibodies may be part of the explanation [13]. Recently, our group reported that a single injection of fremanezumab into rats lowered the basal and capsaicin-provoked CGRP release from the dura mater for up to 30 days, accompanied by reduced blood flow [14] and aversive behaviour to noxious mechanical and thermal facial stimuli [15].

CGRP receptors are heteromers, composed of a seven-transmembrane-spanning protein, the calcitonin receptor-like receptor (CLR), and a one-transmembrane-spanning protein, the receptor-activity-modifying protein 1 (RAMP1) [16]. In addition, an intracellular component, the receptor component protein (RCP), links the membrane components to the intracellular signal transduction, i.e., the dissociation of a Gα_s_-protein, the increase in cAMP and the activation of protein kinase A [17]. RAMP proteins facilitate trafficking of CGRP receptor components and define the ligand specificity of the calcitonin receptor family [18]. The CGRP receptor components RAMP1 and CLR have been identified mainly in medium-sized trigeminal ganglion neurons and satellite glial cells of the trigeminal ganglion, while the CGRP-expressing neurons are smaller on average; importantly, CGRP-producing neurons are different from those expressing CGRP receptors [19,20]. Therefore, CGRP may not only modify the processing of nociceptive information through activation of CGRP receptors, it may also regulate CGRP receptor expression dependent on ambient CGRP levels.

In the current study, we assessed the effects of fremanezumab on plastic changes in primary trigeminal afferents in rats, focusing on alterations in CGRP content and concentration in the trigeminal ganglion, and the immunoreactivity of CGRP and CGRP receptors as measures of changes in CGRP signalling.

## 2. Results

### 2.1. CGRP Plasma Concentration

To collect evidence that a fremanezumab injection was effective in CGRP neutralization, we measured the CGRP plasma concentration in eight animals (four females, four males, later used for immunohistochemistry) 9 or 11 days after the injection of fremanezumab (*n* = 4) or a control antibody (*n* = 4) to equal numbers of males and females. The CGRP concentration in two fractions of each plasma was measured twice and the data were averaged. The CGRP plasma concentration was 30.5 ± 17.1 pg/mL in animals treated with an isotype control antibody vs. 24.9 ± 10.9 pg/mL in animals treated with fremanezumab. The difference was not significant (Mann–Whitney U test, *p* = 0.34).

### 2.2. Body Weight of Animals

Trigeminal ganglia were harvested from 24 rats (12 females and 12 males), either 1, 3 or 10 days after control antibody or fremanezumab administration. The body weight of the animals was statistically compared with factorial ANOVA regarding the factors sex and antibody. Both at the beginning and at the end of the waiting time, females had lower body weight than males (F_1,12_ = 33.72 and 34.21, *p* < 0.0001) but there was no significant difference between animals having received the control antibody versus fremanezumab (F_1,12_ = 0.96 and 0.68, *p* = 0.346 and 0.458).

### 2.3. Trigeminal Ganglion Mass

The mass of the excised ganglia ranged from 9.3 to 29.9 mg (mean ± SD: 16.4 ± 4.9 mg). The ganglia varied in length and attached connective tissue, although care was taken to entirely remove the outer layer of the dura mater. The ganglion mass was not correlated with the body weight of animals (product–moment correlation, *r* = 0.18, *p* = 0.226). Factorial ANOVA showed no significant difference in ganglion mass between left and right ganglia (F_1,24_ = 0.09, *p* = 0.770), between female and male animals (F_1,24_ = 0.95, *p* = 0.339), between fremanezumab and isotype control antibody (F_1,24_ = 3.84, *p* = 0.062) and between the waiting days after antibody injection (F_1,24_ = 0.49, *p* = 0.616).

### 2.4. CGRP Content and Concentration of Trigeminal Ganglia

To minimize the possible bias in CGRP content due to different proportions of connective tissue adhering to the ganglia, we determined the CGRP content extracted from each ganglion (in ng/mL suspension) and calculated the CGRP concentration related to the ganglion weight (in ng/mg ganglion mass).

#### 2.4.1. CGRP Content of Trigeminal Ganglia

The mean CGRP content measured in all trigeminal ganglia was 14.5 ± 4.9 ng/mL with a broad variation (4.4–23.4 ng/mL). Ganglia of female animals contained 11.9 ± 4.2 ng/mL, while those of male animals contained 17.1 ± 4.1 ng/mL. Factorial ANOVA using the factors antibody, sex and day after antibody treatment showed a significant effect between sexes (F_1,36_ = 21.36, *p* < 0.0001). There was also a weakly significant interaction of the factors antibody, sex and day (F_2,40_ = 4.98, *p* < 0.05). Post hoc testing with the Tukey HSD test indicated no significant difference between the specific groups (Figure 1A,B).

#### 2.4.2. CGRP Concentration in Trigeminal Ganglia

The mean CGRP concentration calculated for all trigeminal ganglia was 1.0 ± 0.5 ng/mg with a broad variation (0.4–1.8 ng/mg). Ganglia of female animals contained 0.8 ± 0.5 ng/mg, those of male animals contained 1.1 ± 0.4 ng/mg. Factorial ANOVA using the factors antibody, sex and day after antibody treatment showed a weakly significant effect between sexes (F_1,36_ = 5.71, *p* < 0.05). There was also a weakly significant interaction of the factors antibody and day (F_2,36_ = 3.91, *p* < 0.05) and of the factors sex, antibody and day (F_2,36_ = 5.20, *p* < 0.05). Post hoc testing with the Tukey HSD test indicated no significant difference between the specific groups (Figure 1C,D).

#### 2.4.3. Correlation of CGRP Content and Concentration

The CGRP content of ganglia was not correlated with the ganglion mass (product–moment correlation, *r* = 0.07, *p* = 0.461) but with the CGRP concentration (*r* = 0.77, *p* < 0.0001) (Figure 2). However, there was a negative correlation between the CGRP concentration and the ganglion mass (*r* = −0.66, *p* > 0.0001), probably due to the different amounts of peripheral and central processes with nerve fibres, which contain less CGRP than the cell bodies, and the different amounts of extracellular fluid stored in the ganglia. Therefore, both the CGRP content (ng/mL) and the CGRP concentration per ganglion weight (ng/mg) were used for further calculations.

### 2.5. Immunofluorescence of CGRP and CGRP Receptor Components

The immunofluorescence of CGRP and the CGRP receptor components receptor activity modifying protein 1 (RAMP1), calcitonin receptor-like receptor (CLR) and receptor component protein (RCP) was examined with single and double staining in trigeminal ganglion sections of 14 rats, 10 females and 4 males. Equal numbers of animals were pretreated either with the antibody isotype or with fremanezumab 9–10 days (four females, four males) or 28–30 days (six females) prior to perfusion. Sections from one trigeminal ganglion of each animal were obtained and subjected to immunohistochemical staining. Micrographs were taken from equal numbers of the trigeminal ganglion partitions V1, V2 and V3. Neurons that were immunoreactive to CGRP were counted in micrographs of 94 ganglion sections, for RAMP1 in 68 sections, for CLR in 46 sections and for RCP in 26 sections. Double staining (RAMP1/RCP, CLR/RCP or combinations of CGRP with RAMP1 or CLR) present in 39 sections was analysed using separated channels (Figure 3). Due to the difficulty in distinguishing intensity in double immunofluorescence staining (Figure 3H), we primarily relied on single staining or separated channels for quantification and did not systematically analyse neurons with double immunofluorescence. Micrographs of 43 sections without first antibodies against CGRP or CGRP receptor components were evaluated as negative controls (example in Figure 3C). The average number of neurons counted in each micrograph was 57.2 ± 10.7.

#### 2.5.1. Comparison of CGRP Immunoreactivity

The ratio of all (i.e., weakly and strongly stained) CGRP immunoreactive neurons to all counted neurons was 0.55 ± 0.02 in trigeminal ganglion sections of animals with the isotype antibody and 0.49 ± 0.02 in sections of animals treated with fremanezumab (Figure 4). Using factorial ANOVA with the factor antibody alternatively combined with the factors sex, ganglion division and days after antibody treatment, only the difference between the antibodies was just significant (F_1,90_ = 5.26–6.24, *p* = 0.014–0.024). The post hoc unequal *n* HSD test did not show any significant combination, indicating that this effect is only valid for the whole group of control antibody vs. fremanezumab neurons, so that further differentiation appeared not meaningful.

Analysing only the strongly CGRP immunopositive neurons, apart from a significant difference between the two antibodies (F_1,90_ = 5.98–8.84, *p* = 0.004–0.016), the difference between the two sexes and the two time intervals after antibody injection was also significant (F_1,90_ = 8.20, *p* = 0.005 and F_1,90_ = 10.17, *p* = 0.002); however, the unequal *n* HSD test did not indicate significance between any of the fremanezumab groups, similar to the analysis of all CGRP immunopositive neurons.

#### 2.5.2. Comparison of CGRP Receptor Immunoreactivity

The ratio of all RAMP1-immunopositive neurons to all counted neurons was 0.57 ± 0.02 in trigeminal ganglion sections of animals treated with the isotype control antibody and 0.37 ± 0.04 in sections of animals treated with fremanezumab (Figure 5). Data were compared using factorial ANOVA with the factor antibody, alternatively combined with the factors trigeminal ganglion division, sex and days after antibody treatment. In all these comparisons, the difference between control antibody and fremanezumab was highly significant (F_1,62/64_ = 21.40–23.68, *p* < 0.0005) but there was no difference between the three trigeminal partitions V1-V3 (F_2,62_ = 0.80, *p* = 0.45), between females and males (F_1,64_ = 0.65, *p* = 0.42) and between different waiting times (F_1,64_ = 0.64, *p* = 0.43). Post hoc testing using the unequal *n* HSD test confirmed that the difference between the two antibody treatments was significant in both sexes and independent on the waiting time, but with a higher significance in females and after 10 days (*p* < 0.005) than in males and after 30 days (*p* < 0.05), whereas it was only significant in trigeminal ganglion divisions V2 (*p* < 0.05) and V3 (*p* < 0.005).

The ratio of all CLR-positive neurons was 0.56 ± 0.04 in control animals and 0.33 ± 0.04 in fremanezumab-treated animals (Figure 5). This difference was also significant (F_1,40/42_ = 9.14–13.60, *p* < 0.005), and again, without difference between trigeminal divisions (F_2,40_ = 2.73, *p* = 0.08), sexes (F_1,42_ = 0.04, *p* = 0.84) and waiting time (F_1,42_ = 0.23, *p* = 0.63). Post hoc testing showed that the difference was dependent on males and the waiting time of 10 days (*p* < 0.05).

The ratio of all RCP-positive neurons was 0.62 ± 0.15 in control animals and 0.60 ± 0.05 in fremanezumab-treated animals (Figure 5), which is statistically not different (F_1,19_ = 0.47, *p* = 0.50). However, there was a just significant difference between the trigeminal divisions (F_2,19_ = 5.13, *p* = 0.02). Since there was no difference in RCP between the antibodies, the ratio of double-stained RCP neurons with RAMP1 or CLR was not calculated. Similar ratios to those in all immunoreactive neurons resulted from a separate counting of weakly and strongly immunopositive neurons (Figure 5).

## 3. Discussion

Monoclonal antibodies targeting CGRP are currently used for the prevention of chronic and frequent episodic migraine [3,21]. The antibodies target CGRP, which may be released from activated primary afferents, but their mechanism of action is not entirely clear. The underlying mechanism through which CGRP activates trigeminal afferents is yet to be fully elucidated and could potentially involve a cycle of CGRP release and subsequent amplification of afferent activity [22,23]. This detrimental cycle could significantly contribute to the pathophysiological processes involved in migraine pain generation. Since the nociceptive processes underlying headache generation are similar in all mammals, preclinical animal experiments focusing on the CGRP signalling system are frequently used to model migraine generation.

### 3.1. Impact of Fremanezumab on CGRP and CGRP Receptor Presence

The current experimental data show that up to 30 days after a single injection of the monoclonal anti-CGRP antibody fremanezumab, the number of trigeminal ganglion neurons immunoreactive to CGRP and the CGRP receptor components, RAMP1 and CLR, but not RCP, were reduced compared to an isotype control antibody. However, the same fremanezumab treatment had no consistent and significant impact on the CGRP content and concentration in the trigeminal ganglion. This discrepancy may be the result of the different methods used. All experiments, from antibody injection to cell counting, were conducted under strictly blinded conditions, but we are aware of the limited validity of quantifying cells marked by immunofluorescence and the confounding factors of the ELISA methods used. Nevertheless, in situ immunofluorescence may be more sensitive to changes in CGRP presentation and less prone to interferences than the ELISA, which was performed after cell lysis and peptide separation. Also, differences depending on sex did not significantly influence the immunohistochemical results but had a major impact on CGRP content and concentration (see Section 3.2).

We found significant downregulation of the immunoreactivity for the CGRP receptor components RAMP1 and CLR after fremanezumab treatment but only a small reduction in CGRP immunoreactive neurons. This specific regulation is not surprising, because CGRP receptors are expressed by neurons that do not express CGRP and vice versa [19,20], so that only neurons with CGRP receptors can directly be influenced by ambient CGRP. Therefore, assuming that CGRP receptor expression is regulated by the available CGRP concentration, we regard a direct CGRP effect as most likely. It is important to consider that in addition to trigeminal neurons and vascular smooth muscle cells, CGRP receptors can also be expressed by glial cells (Schwann cells and satellite cells in the trigeminal ganglion) and mononuclear cells such as mast cells [19,24]. Being aware of the methodological limits of this study, we hypothesise that the long-lasting effect of fremanezumab disrupts the CGRP cross-signalling between trigeminal ganglion neurons [23], slowing down the synthesis or allocation of CGRP receptors in trigeminal neurons.

The maximal effect after fremanezumab injection regarding changes in CGRP release from the dura mater was seen after 10 days [14]. Therefore, we have primarily chosen this period for the experiments of the current study. Speculating that changes in expression pattern are even more delayed, we have added the 30-day period. At this time, there was still a significant effect of lowering CGRP release after fremanezumab [14].

Interestingly, only the CGRP receptor components RAMP1 and CLR appeared to be downregulated after fremanezumab, but this was not the case for RCP. RAMP1 is not only an element of the canonical CGRP receptor but also part of the amylin-1 receptor, the second receptor with a high affinity for CGRP [25]. Therefore, it is possible that part of the RAMP1 immunofluorescence is indicating amylin-1 receptors [24]. Also, CLR is a component not only of CGRP, but also of adrenomedullin receptors, and therefore not exclusively indicative for CGRP receptors [26]. However, in the trigeminal ganglion, adrenomedullin receptors are mainly found to be associated with satellite glial cells [24]. The observation that RCP, which contributes to intracellular CGRP signalling [27], did not change after fremanezumab treatment is an interesting detail but is not contradictory to the selective downregulation of RAMP1 and CLR receptor components.

### 3.2. Sex Difference in CGRP Content and Concentration

Significant sex differences were observed in CGRP content and concentration within the trigeminal ganglion, with notably higher levels found in male rats compared to female rats. However, no significant differences were observed in CGRP immunoreactivity. The relevance of this difference is not clear. The effect cannot be explained by a lower ganglion mass, since no sex difference was found there. The higher CGRP in male trigeminal ganglia is the more remarkable, as our group has previously found that the basal and the capsaicin-stimulated CGRP release from the rat cranial dura mater is higher in females compared to males [14]. Also, according to this previous data, fremanezumab reduced the CGRP release in female rats more robustly than in male rats. In addition, behavioural signs of increased facial sensitivity to unpleasant mechanical and heat stimuli were reduced only in female, but not male, animals treated with fremanezumab [15]. Therefore, we increased the number of female animals used for immunohistochemistry but found no significant sex difference in CGRP and CGRP receptor immunofluorescence. It would be interesting to look at the density of the CGRPergic afferent innervation of the dura mater, comparing male rats and female rats. However, the available data indicate that functional rather than structural differences underlie the greater sensitivity of female animals to CGRP and their higher susceptibility to inhibition of CGRP signalling, respectively [28].

In view of the clinical relevance, the current results are interesting because of the well-known significantly higher incidence of migraine in women, which can only partly be explained by changes in hormonal levels [29,30,31]. The higher susceptibility of migraine-promoting factors in females is currently an important issue in preclinical experiments determining differences in transduction mechanisms, on which the release of neuropeptides like CGRP directly depend [32,33,34]. Also, application of CGRP on the dura mater has been found to cause female-specific migraine-like responses in rodents [35]. However, the pathophysiological mechanisms linking these findings are widely unclear.

### 3.3. Conclusions

The present results suggest that the monoclonal anti-CGRP antibody fremanezumab has long-lasting effects on CGRP signalling in rats not only by reducing CGRP but particularly by lowering the expression of the CGRP receptor components RAMP1 and CLR, thus decreasing the availability of CGRP receptors in trigeminal ganglion neurons. A clear effect of fremanezumab on the CGRP production could not be deduced from the data. We postulate that the reduction in CGRP receptor components may be one of the reasons for the antinociceptive effect of fremanezumab.

## 4. Materials and Methods

Animal housing and all experiments were carried out according to the German guidelines and regulations on the care and treatment of laboratory animals and the European Communities Council Directive of 24 November 1986 (86/609/EEC), amended 22 September 2010 (2010/63/EU). The experimental protocols were reviewed by an ethics committee and approved by the District Government of Middle Franconia (54-2532.1-21/12).

### 4.1. Animals

Adult Wistar rats of both sexes (age 2–4 months; body weight of 22 females: 230–370 g; 16 males: 230–490 g), bred and housed in the animal facility of the Institute of Physiology and Pathophysiology of the FAU Erlangen-Nürnberg, were used. They were kept at a 12 h light/dark cycle in standard cages in groups of 3–4, fed with standard food pellets and water ad libitum and supplied with material for behavioural enrichment. Animals were matched and distributed according to their sex and weight as equally as possible for the experiments. The oestrus state of females was not assessed.

### 4.2. Administration of Antibodies

Rats were anaesthetised around 9 a.m. in a plastic box with isoflurane (Forene, Abott, Wiesbaden, Germany) at a rising concentration up to 4% using an evaporator (Forane Vapor 19.3, Dräger AG, Lübeck, Germany). Animals were weighed, individually marked at the tail and the neck region was shaved and disinfected with 70% ethanol. Then, using a syringe with a 27-gauge needle, either fremanezumab or isotype control antibody (30 mg/kg) was subcutaneously injected, evenly distributed 2 cm left and right from the midline and 5 cm caudal of the occiput. As active anti-CGRP antibody, fremanezumab (Teva Pharmaceuticals, Redwood City, CA, USA) diluted in saline (10 mg/mL) was used, alternatively a human IgG2 antibody (Teva Pharmaceuticals) targeting keyhole limpet hemocyanin (KLH) as the isotype control antibody, to demonstrate that effects seen only in the fremanezumab group are specifically due to targeting CGRP, was used. The examiners were blinded to the identity of the antibodies. Animals were placed back in their cage, where they recovered from anaesthesia usually within 2–3 min. During the following days, they were inspected every day to register any unusual behaviour. Body weight of the animals was measured 1, 3 or 10 days after control antibody or fremanezumab administration.

### 4.3. Preparation for CGRP Measurements

On day 1, 3 or 10 after antibody injection, rats (12 females, 12 males) were deeply anaesthetized and sacrificed in a rising CO_2_ atmosphere (Figure 6A). The head was separated, skinned and symmetrically cut into halves along the sagittal line. The trigeminal ganglia were excised from the skull base at a length of 6–7 mm, placed in Eppendorf cups and frozen at −20 °C until further processing.

For measurement of CGRP content, the cryoconserved samples were thawed, tipped on an absorbent paper to remove the adhering fluid and weighed. Then, they were immersed in 2 mL 2 M acetic acid, heated to 95 °C and boiled for 10 min in glass tubes. Afterwards, the tissues were homogenized with a custom-built disperser, boiled a second time for 10 min and centrifuged for 10 min at 2000 rpm. The supernatant was collected; the pH was around 6, which is suitable for the CGRP analysis.

### 4.4. Analysis of CGRP Concentration

Samples (100 µL) of the collected fluid were separated, 25 µL of enzyme-immunoassay (EIA) buffer containing peptidase inhibitors (Bertin Pharma/SPIbio, Montigny le Bretonneux, France) was added and samples were processed using an EIA kit for CGRP according to the instructions of the manufacturer (Bertin Pharma/SPIbio). The EIA is based on a double-antibody sandwich technique with monoclonal capture and tracer antibodies binding the CGRP molecule; the tracer antibody is conjugated with acetylcholine esterase converting Ellman’s reagent to a yellow substance, the absorbance of which is measured by a photo-spectrometer (Opsys MR, Dynex Technologies, Denkendorf, Germany). The assay has 100% reactivity to rat CGRP but < 0.01% cross-reactivity to other proteins of the calcitonin family and detects both α- and β-CGRP with the same sensitivity. The lower limit of detection is 2 pg/mL according to the manufacturer’s information. The CGRP content of trigeminal ganglia was calculated in pg/mL, considering the added volume of EIA buffer. The relative CGRP concentration in each of the trigeminal ganglia was calculated by dividing the CGRP content in the samples by the ganglion weight (pg/mg).

Statistical analysis was performed on non-normalized values using Statistica (StatSoft, Tulsa, OK, USA). Mann–Whitney U test and product–moment correlation were applied. Following verification of normal distribution of data, analysis of variance (factorial ANOVA) was used, extended by the Tukey’s honest significant difference (HSD) test, as specified in the results. The level of significance was set at *p* < 0.05. Data are displayed as mean ± standard deviation (SD).

### 4.5. Preparation for Immunohistochemistry

Ten or thirty days after administration of fremanezumab or control antibody, respectively (Figure 6B), animals (10 females, 4 males) were anaesthetized with isoflurane as described and euthanized with an intraperitoneal overdose of thiopental-Na (Trapanal, Sigma-Aldrich, Taufkirchen, Germany). Immediately after cessation of the spontaneous ventilation, animals were quickly thoracotomized. From 8 animals (4 treated with fremanezumab and 4 with control antibody), blood was drawn from the left ventricle with a syringe and centrifuged at 3000 rpm for 15 min. The plasma was collected, EIA buffer (1:4) was added and plasma was frozen (see above). All animals were perfused through the left ventricle with warm isotonic saline for about 2 min, followed by a solution of 4% paraformaldehyde in 0.1 M phosphate buffer (pH 7.4) for 6–10 min. Following craniotomy, the trigeminal ganglia were carefully dissected, washed in phosphate-buffered saline (PBS, pH 7.4) overnight and stored one day in 20% buffered sucrose for cryoprotection. The ganglia were mounted on Tissue-Tek (Sakura Finetek NL, Science Service, München, Germany), rapidly frozen in methylbutane at −46 °C and stored at −20 °C. Using a cryostat (Leica Mikrosysteme, Bensheim, Germany), series of 15 µm thick sections were cut in the horizontal plane, mounted on poly-L-lysine-coated slides and dried for 1 h at room temperature.

### 4.6. Immunohistochemistry

From each ganglion (i.e., each animal), 2 slides with the 12 most central sections (showing a maximal number of neurons in all three trigeminal branches) were selected and incubated with 5% normal goat serum containing 0.5% Triton X-100 (Merck, Darmstadt, Germany) and 1% bovine serum albumine (BSA) in PBS for 1 h at room temperature. Thereafter, they were rinsed in PBS and incubated for immunolabeling with appropriate polyclonal primary antibodies raised against CGRP or the CGRP receptor components, RAMP1, CLR or RCP, in PBS with 1% BSA and 0.5% Triton X-100 at room temperature overnight. Two sections of each slide were used for single staining, two for double staining with a specific combination of two of the target proteins and two for control of the secondary antibodies without first antibodies. For the specification of antibodies used, see Table 1. After another wash, the sections were incubated with the second antibodies, IgG directed against the species, in which the first antibody was raised and dissolved in PBS with 0.5% Triton X-100 and 1% BSA at room temperature (see Table 1). After a further rinse, the sections were mounted with Roti-Mount FluorCare DAPI (4′,6′-diamidino-2-phenylindole hydrochloride; Sigma Aldrich, Taufkirchen, Germany) and coverslipped. Specificity of the immunohistochemical reactions was tested by omission of primary antibodies, which were replaced by PBS with 1% BSA and 0.5% Triton X-100.

### 4.7. Imaging

Confocal imaging was performed with an inverse stage LSM 710 Axio Examiner Z1 microscope (Carl Zeiss) using dry objective lenses at magnifications of 10, 20 and 40 and appropriate filter settings of the confocal scanner. First, the optical mode was used to focus the image and to select one or two regions with the maximum of visible neurons in each of the three trigeminal divisions (V1–V3). Thus, the selected regions were approximately matching in all sections. After switching to the laser scanning mode, pinhole settings and amplification were adjusted for each channel to an optimal range and focus. The same adjustment was used for the whole section series with the same type of immunostaining and the control staining without antibodies. Images of 512 × 512 pixels were obtained from the selected regions at a magnification of 200. The channels of each image were merged into a 12-bit RGB tiff-file using confocal assistant software ZEN 2010 (Carl Zeiss, Oberkochen, Germany).

The same software was used to evaluate the images and to visually count the number of neurons within the selected regions. For counting all, including unstained, neurons, the range of grey values from black (0) to white (255) was compressed to 0–160 enhancing saturation, so that the background staining was also visible. For counting immunopositive neurons in each channel and, in the case of double staining, in the overlain channels, the range of grey values was restricted, cutting lower grey values and thereby enhancing the saturation of higher values, as follows. For counting all immunopositive neurons, the scale was compressed from 30 to 150, and at a compression of 60–150, only strongly positive neurons appeared (Figure 7). For counting CGRP immunopositive neurons, we used an additional criterion. Neurons appearing as “filled” with immunoreactivity were regarded as strongly immunopositive, those appearing with granulated or partial staining were taken as weakly positive. All neurons with a visible nucleus were counted. At the borders of the field of vision, neurons were counted if more than half of them and a complete nucleus were visible. The immunostaining was evaluated by the same examiner, blinded for the experiment.

For calculating the ratio of immunopositive neurons, strongly and weakly positive neurons were divided through the whole number of neurons counted in each micrograph. The cell ratios of all micrographs were analysed using Statistica (StatSoft, Tulsa, OK, USA). Factorial ANOVA extended by the unequal N HSD test was used to differentiate the results according to the factors antibody and sex or antibody and trigeminal divisions (V1–V3). The level of significance was set at *p* < 0.05. Data were presented as means ± standard deviation (SD).

## Figures and Tables

**Figure 1 ijms-24-13471-f001:**
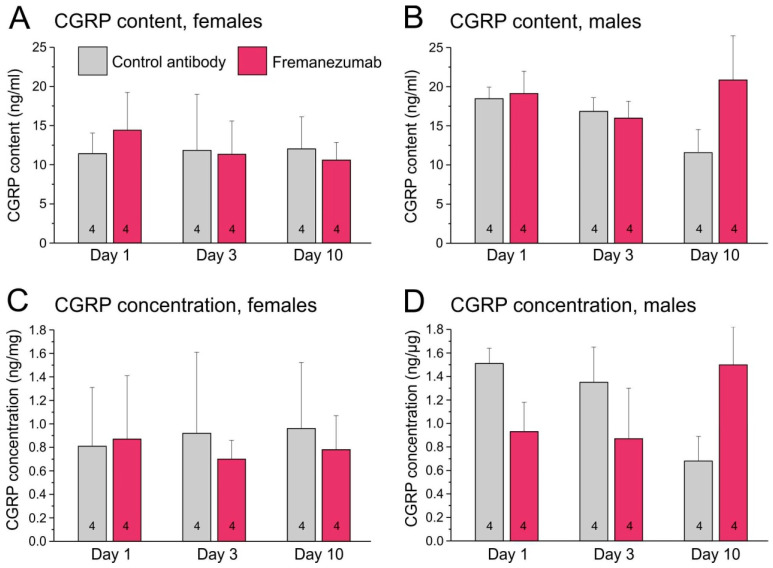
Calcitonin gene-related peptide (CGRP) content (**A**,**B**) and concentration (**C**,**D**) in trigeminal ganglia of female (**A**,**C**) and male rats (**B**,**D**). Numbers of ganglia are indicated in the bars.

**Figure 2 ijms-24-13471-f002:**
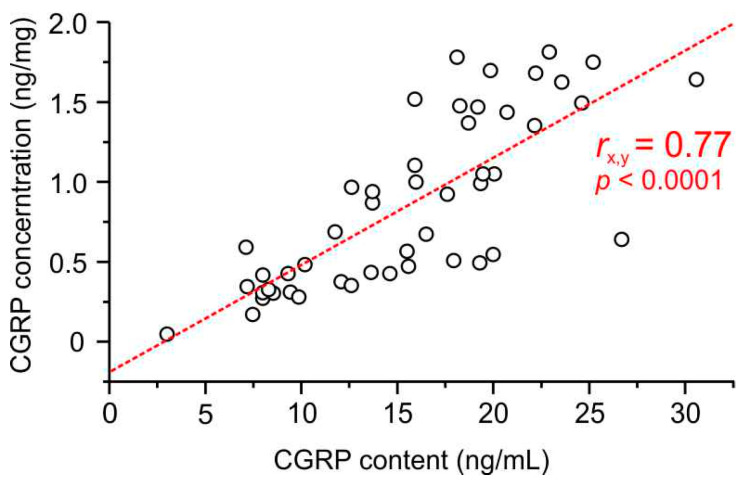
Correlation of the measured CGRP content and the calculated CGRP concentration in 48 trigeminal ganglia (*r*_x,y_, correlation coefficient).

**Figure 3 ijms-24-13471-f003:**
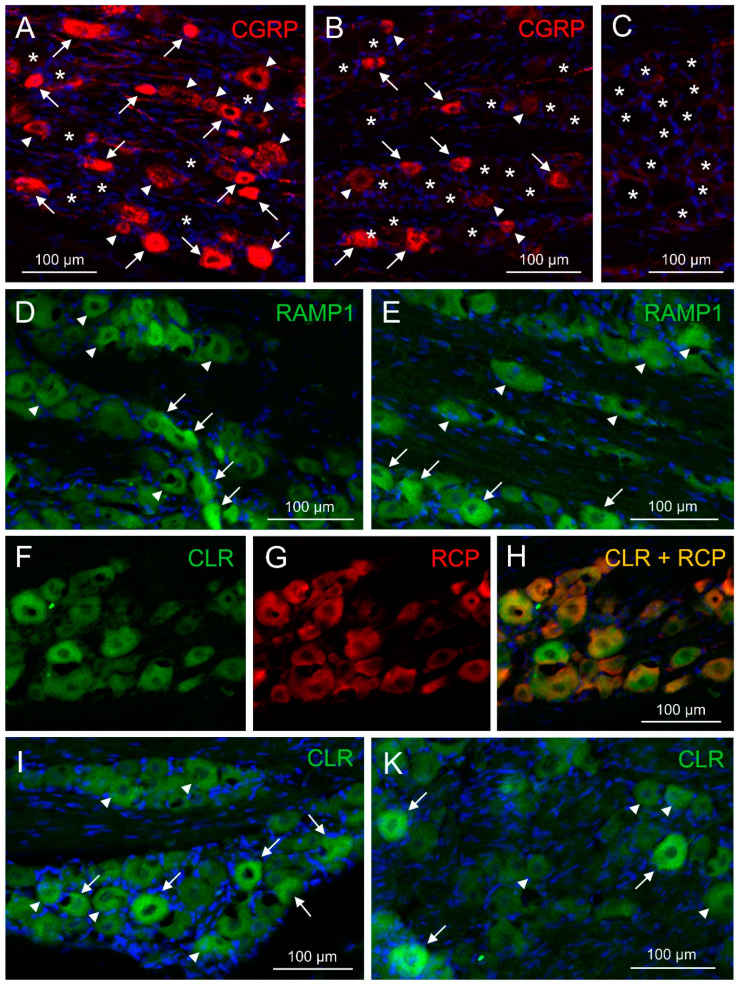
Typical examples of corresponding images used for cell counting of neurons immunopositive for CGRP, receptor activity modifying protein 1 (RAMP1), calcitonin receptor-like receptor (CLR) and receptor component protein (RCP) in the V1 region (**D**,**E**) and the V2 region (**A**–**C**,**F**–**I**,**K**). (**A**,**D**,**I**) represent ganglion sections of animals treated with control antibody, (**B**,**E**,**K**) are from animals treated with fremanezumab, (**C**) shows part of a control section without primary antibody. (**H**) is an overlay of (**F**,**G**) showing co-localization of CLR and RCP immunofluorescence. Arrows point to neurons classified as strongly immunopositive, arrowheads mark exemplary neurons regarded as weakly positive; stars in (**A**–**C**) show unstained neurons, which are only visible in high brightness or indicated by their surrounding satellite cells (blue DAPI staining). Red fluorescence is produced by a secondary antibody conjugated with Alexa 555, green fluorescence with Alexa 488.

**Figure 4 ijms-24-13471-f004:**
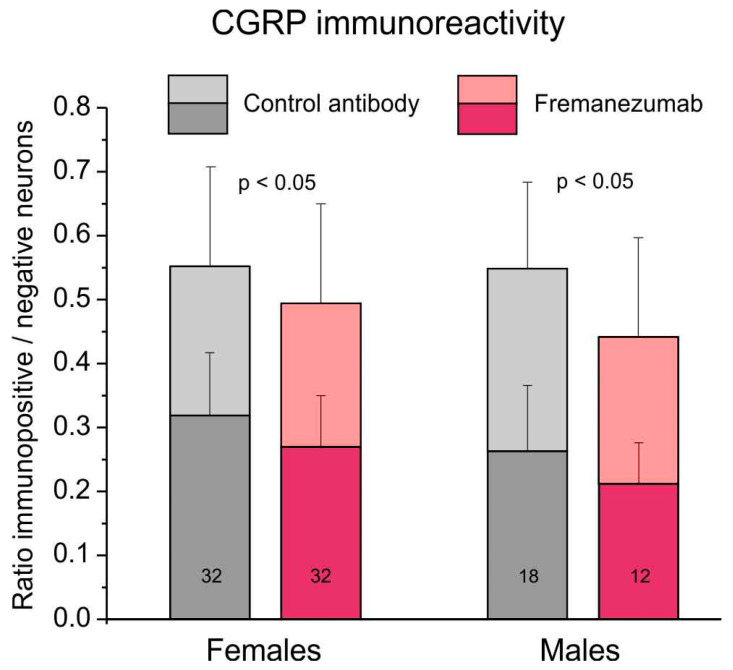
Fraction of the number of neurons immunoreactive to CGRP in trigeminal ganglia of rats treated with control antibody versus fremanezumab. The dark colours indicate strongly immunopositive, and the light colours weakly, immunopositive neurons. The numbers of sections used for counting are indicated in the bars.

**Figure 5 ijms-24-13471-f005:**
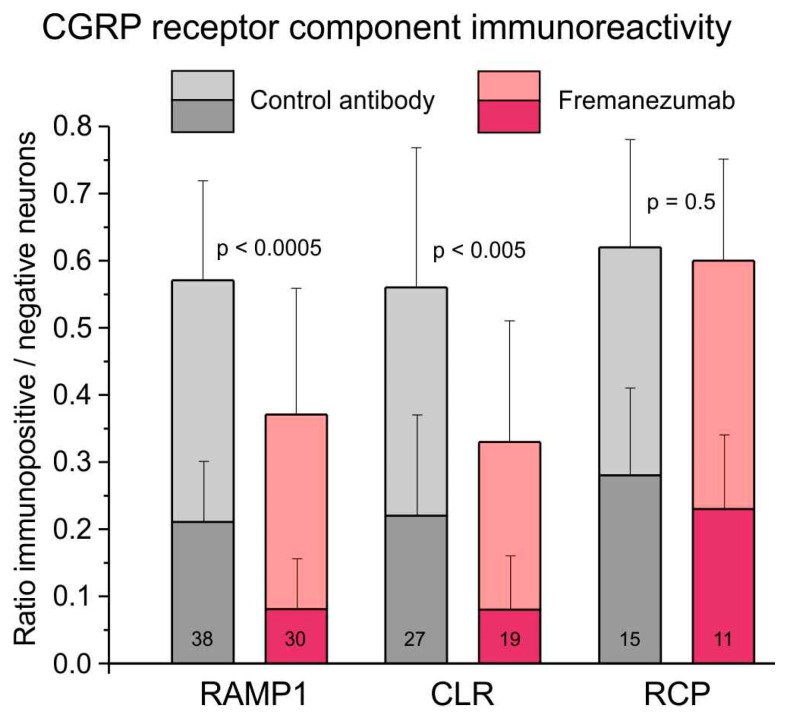
Fraction of the number of neurons immunoreactive to the CGRP receptor components CLR, RAMP1 and RCP in trigeminal ganglia of rats treated with control antibody versus fremanezumab. The dark colours indicate strongly immunopositive and the light colours weakly immunopositive neurons. The numbers of sections used for counting are indicated in the bars.

**Figure 6 ijms-24-13471-f006:**
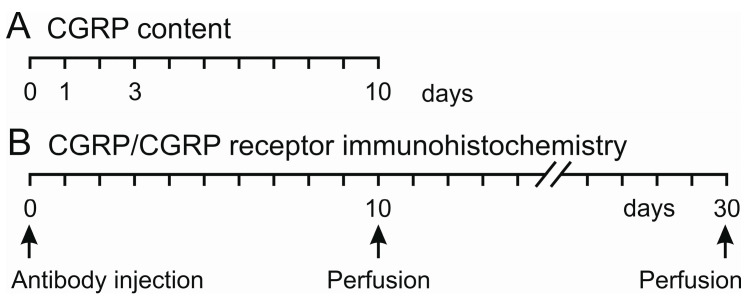
Experimental timeline for measurements of CGRP content (**A**) and preparation for immunohistochemistry (**B**) after injection of fremanezumab or isotype control antibody at day 0.

**Figure 7 ijms-24-13471-f007:**
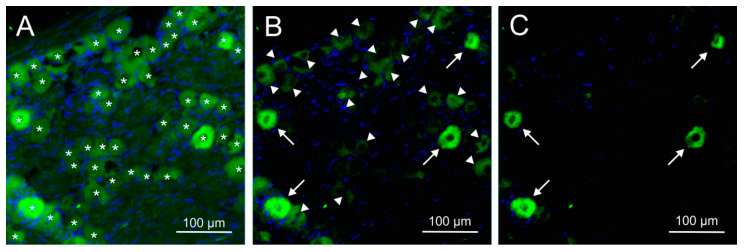
Method for counting and differentiating immunoreactive neurons. Only neurons with a clearly visible nucleus were counted; at the borders of the image, neurons were counted if at least half of them were visible. This example of CLR immunostaining is underlying Figure 3K. (**A**) Counting of all neurons. The original grey value ranging from 0 (black) to 255 (white) was compressed to 0–150, enhancing the intensity to show also the background staining; * denote all counted neurons. (**B**) Selective counting of immunopositive neurons. The range was restricted to 30–150, weakly immunopositive neurons are marked with arrowheads. (**C**) Counting of strongly immunopositive neurons (arrows) after further compression of the grey value range to 60–150.

**Table 1 ijms-24-13471-t001:** List of antibodies.

Primary Antibodies				
Target/Epitope	Host Species	Manufacturer	Specification	Dilution
Rabbit α-CGRP	Rabbit	BMA Biomed., Augst, Switzerland www.bma.ch; last access Aug 2023	T4032 Peninsula	1:1000
Rat α-CGRP, β-CGRP, monoclonal	Mouse	Santa Cruz Biotechnology, Heidelberg, Germany www.scbt.com; last access Aug 2023	Sc-57053/CGRP (4901)	1:10
Human CLR (C-terminus), polyclonal	Goat	Santa Cruz Biotech.	CRLR (V-20): sc-18007	1:10
Human RAMP1 (aa 27–117), monoclonal	Mouse	Santa Cruz Biotech.	RAMP1 (3B9): sc-293438	1:10
Human/mouse RAMP1 (aa 1–148), polyclonal	Rabbit	Santa Cruz Biotech.	RAMP1 (FL-148): sc-11379	1:10
Human/mouse RAMP1 (C-terminus), polyclonal	Goat	Santa Cruz Biotech.	RAMP1 (C-20): sc-8851	1:10
RCP, monoclonal	Mouse	Santa Cruz Biotech.	CGRP-RCP: sc-393347	1:10
**Secondary antibody conjugated to**				
Goat anti-rabbit Cy3		Dianova, Hamburg, Germany www.dianova.com; last access Aug 2023	111-164-144	1:100
Donkey anti-mouse Alexa Fluor 555		Invitrogen, Bonn, Germany www.thermofisher.com; last access Aug 2023	A 31 570	1:500
Donkey anti-rabbit Alexa Fluor 488		Molecular Probes, Bonn, Germany www.thermofisher.com; last access Aug 2023	A 21206	1:500
Donkey anti-goat Alexa Fluor 488		Molecular Probes	A 11055	1:500

## Data Availability

Not applicable.

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
