# Peer review of "The Anti-Calcitonin Gene-Related Peptide (Anti-CGRP) Antibody Fremanezumab Reduces Trigeminal Neurons Immunoreactive to CGRP and CGRP Receptor Components in Rats"

_ijms, 2023, doi:10.3390/ijms241713471_

Round 1
Reviewer 1 Report
Comments to the Author
The manuscript by Vogler and colleagues investigated the effects of monoclonal anti-CGRP antibody fremanezumab on CGRP and CGRP receptor components expressed in the trigeminal ganglion. Fremanezumab reduced the expression of CGRP receptor while the concentration of CGRP in trigeminal ganglia remained unaltered. The authors were tempted to translate the present findings into a possible explanation for the long-lasting antinociceptive effect after a single administration of monoclonal anti-CGRP antibodies. The study is interesting and very well written.
Minor concerns.
Please replace Gs-protein with Gα -proteins (page 2).
How is it possible to measure the number of neurons immunoreactive for CGRP upon administration of antibodies neutralization CGRP molecules?
A section in the result about “Sex difference in CGRP content and concentration” is missing.
The mentioned cycle of CGRP release and subsequent amplification needs more elaboration? (page 2).
The logical expectation upon targeting and neutralization CGRP molecules is that CGRP concentration increases to replace neutralized CGRP molecules… Any comment? The same expectation goes for CGRP receptors.
The authors reported that the number of neurons immunoreactive for CGRP and the CGRP receptor components are reduced after fremanezumab. But how about the expression of CGRP receptor within the same neurons before and after fremanezumab?
Kind regards
Mohammad Al-Mahdi Al-Karagholi
Author Response
Comments to the Author
The manuscript by Vogler and colleagues investigated the effects of monoclonal anti-CGRP antibody fremanezumab on CGRP and CGRP receptor components expressed in the trigeminal ganglion. Fremanezumab reduced the expression of CGRP receptor while the concentration of CGRP in trigeminal ganglia remained unaltered. The authors were tempted to translate the present findings into a possible explanation for the long-lasting antinociceptive effect after a single administration of monoclonal anti-CGRP antibodies. The study is interesting and very well written.
Thank you very much for your encouraging review.
Minor concerns.
Please replace Gs-protein with Gα -proteins (page 2).
Gs has been replaced by Gas
How is it possible to measure the number of neurons immunoreactive for CGRP upon administration of antibodies neutralization CGRP molecules?
This is a very interesting question. Anti-CGRP antibodies like fremanezumab do not pass the cell membrane but stay in the extracellular compartment. Through the perfusion of the animals prior to fixation but also during intense washing of the histological sections, fremanezumab may be completely washed off. However, given that some fremanezumab is fixed to cell membranes, there may be indeed some competition with the primary antibody recognizing CGRP, if the two antibodies bind to the same epitope of the CGRP molecule. In this case, the difference in immunostaining between fremanezumab and control antibody treated samples would be even more pronounced.
A section in the result about “Sex difference in CGRP content and concentration” is missing.
Since sex differences were not in the focus of the present paper, we have noted this data in the paragraphs about CGRP content and concentration. However, we used a specific paragraph in the Discussion to discuss sex differences regarding CGRP content and concentration (higher in male) in the present paper and CGRP release (higher in females) in previous papers of our group. To complete this discussion, we have now added another sentence to this paragraph: “In addition, behavioural signs of increased facial sensitivity to unpleasant mechanical and heat stimuli were reduced only in female but not male animals treated with fremanezumab [12].”
The mentioned cycle of CGRP release and subsequent amplification needs more elaboration? (page 2).
We have added a sentence to the Introduction, which shortly mentions the basic idea of the cross-activation of primary trigeminal afferents: “In short, the basic idea of these hypotheses is that CGRP released from primary afferents in the meninges or the trigeminal ganglion is activating another type of (not CGRP releasing) afferents directly or via glial cells that produce excitatory substances like nitric oxide”. Since the three hypotheses cited here (refs 7-9) are different in their particular postulated mechanisms, we abstain from explaining them more in detail but added two other references.
The logical expectation upon targeting and neutralization CGRP molecules is that CGRP concentration increases to replace neutralized CGRP molecules… Any comment? The same expectation goes for CGRP receptors.
Yes, good question. This was indeed our first hypothesis but unexpectedly the opposite result was found. Obviously there is no upregulation but rather downregulation of CGRP and CGRP receptors (judged from immunohistochemistry), which underpins the existence of a hypothetic feed-forward loop in CGRP/CGRP receptor expression.
The authors reported that the number of neurons immunoreactive for CGRP and the CGRP receptor components are reduced after fremanezumab. But how about the expression of CGRP receptor within the same neurons before and after fremanezumab?
It would be great to answer this question but this is not possible with the current methods, because the animals must be sacrificed for the CGRP content measurements and immunohistochemistry.
Reviewer 2 Report
In this manuscript, the authors investigated the antinociceptive effects of fremanezumab in rats, using immunohistochemical and ELISA techniques to analyze the content and concentration of CGRP in the trigeminal ganglion, as well as the ratio of trigeminal ganglion neurons immunoreactive with CGRP and CGRP receptor components 1-10 days after fremanezumab after subcutaneous injection, compared with an isotype control antibody.
It was found that after fremanezumab treatment, the fraction of trigeminal neurons immunoreactive for CGRP and CGRP receptor components, calcitonin receptor-like receptor (CLR) and receptor activity modifying protein 1 (RAMP1) decreased significantly compared to the control.
In addition, the content and concentration of CGRP in the trigeminal ganglia did not change significantly.
Based on these, it can be assumed that the long-term reduction of CGRP receptors expressed in trigeminal afferents may contribute to the weakening of CGRP signaling and the antinociceptive effect of monoclonal anti-CGRP antibodies in rats.
The topic is timely and may attract much attention. However, in its current version, the manuscript has several limitations that should be addressed:
Materials and methods:
1. It would be worthwhile to indicate the number of animals used (total and per experiment).
2. The variance in the weight of the animals used is quite large. Although the animals were matched for the experiments according to their sex and weight and distributed as evenly as possible, to what extent was it possible to allocate the animals in such a way that at least for the different experiments animals of the same sex were used of approximately the same weight? Could the difference in weight have affected the results?
3. It would be worth noting how old the animals were.
4. "The oestrus state of females was not assessed." Why? After all, hormonal changes can affect the behavior of animals, the feeling of pain, etc.
5. A timeline would be helpful. It is easier for readers to understand how the experiment took place, when what happened, how did the experiments come one after the other?
6. Was there an animal that was used in more than one experiment?
7. If there was an experiment that was performed 30 days after fremanezumab administration, why were the animals only monitored for the first 10 days?
8. I find 4 animals per group insufficient for immunohistochemical testing.
9. How was it chosen that the CGRP measurement should be done on days 1, 3, and 10, while the immunohistochemistry was done on day 10 or 30 after fremanezumab treatment?
References:
In general, I recommend authors use more references to back their claims. I believe that adding more citations will help to provide better and more accurate background to this study.
I recommend this manuscript for publication after major revision.
Author Response
Comments and Suggestions for Authors
In this manuscript, the authors investigated the antinociceptive effects of fremanezumab in rats, using immunohistochemical and ELISA techniques to analyze the content and concentration of CGRP in the trigeminal ganglion, as well as the ratio of trigeminal ganglion neurons immunoreactive with CGRP and CGRP receptor components 1-10 days after fremanezumab after subcutaneous injection, compared with an isotype control antibody.
It was found that after fremanezumab treatment, the fraction of trigeminal neurons immunoreactive for CGRP and CGRP receptor components, calcitonin receptor-like receptor (CLR) and receptor activity modifying protein 1 (RAMP1) decreased significantly compared to the control.
In addition, the content and concentration of CGRP in the trigeminal ganglia did not change significantly.
Based on these, it can be assumed that the long-term reduction of CGRP receptors expressed in trigeminal afferents may contribute to the weakening of CGRP signaling and the antinociceptive effect of monoclonal anti-CGRP antibodies in rats.
The topic is timely and may attract much attention. However, in its current version, the manuscript has several limitations that should be addressed:
Materials and methods:
- It would be worthwhile to indicate the number of animals used (total and per experiment).
We have used 24 animals (12 males, 12 females) for CGRP content measurement and 14 (4 males, 10 females) for immunofluorescence experiments, altogether 38 animals. The numbers had been mentioned in the Results and have now also been added to the Methods.
- The variance in the weight of the animals used is quite large. Although the animals were matched for the experiments according to their sex and weight and distributed as evenly as possible, to what extent was it possible to allocate the animals in such a way that at least for the different experiments animals of the same sex were used of approximately the same weight? Could the difference in weight have affected the results?
The mean body weight of female animals treated with control antibody was 270 ± 22 g and those treated with fremanezumab was 292 ± 47 g. The mean body weight of male animals was 400 ± 58 g in control antibody and 412 ± 69 g in fremanezumab treated animals, i.e., the difference between control antibody and fremanezumab treated animals was within the standard deviation of the specific groups. Until day 3, animals gained weight (females 5 g, males 7.5 g), until day 10 they gained more weight (females 27.5 g, males 50 g) but there was again no significant difference between control antibody and fremanezumab treatment.
- It would be worth noting how old the animals were.
The age of the animals ranged from 2 to 4 months, most of them were about 3 months old. We have added this information to the Methods.
- "The oestrus state of females was not assessed." Why? After all, hormonal changes can affect the behavior of animals, the feeling of pain, etc.
The reviewer is right in stating that hormonal changes can affect the behaviour of animals. The oestrus cycle of rats last about 4 days with some variability (Robert et al. 2021), whereas the observation time after antibody injection was 1-30 days. The first antibody effect was seen at day 3 after the treatment and was maximal at day 10 (Dux et al. 2022). This means that two or more oestrus cycles passed over the experimental time of 10-30 days, and it is extremely difficult to define a study schedule that considers this complexity. Since the variability of results in female animals was not higher than in males, we abstained from considering the oestrus state.
- A timeline would be helpful. It is easier for readers to understand how the experiment took place, when what happened, how did the experiments come one after the other?
Timeline with legend has been added to the manuscript as Figure 6.
- Was there an animal that was used in more than one experiment?
Each animal was used for only one experiment.
- If there was an experiment that was performed 30 days after fremanezumab administration, why were the animals only monitored for the first 10 days?
Animals were monitored after injection of either antibody until they were used for the experiments, up to 30 days.
- I find 4 animals per group insufficient for immunohistochemical testing.
We have well considered this critical point before analysing the data. The intra-individual variability (i.e., variability of counted immunopositive neurons in the sections of one and the same animal) was comparable with the variability between animals. For example, the computed variance of the ratio of CGRP immunoreactive neurons was 27% over all trigeminal ganglion sections, 21% for fremanezumab as well as for control antibody. However, variance ranged in individual ganglia (= animals) from 2% to 22% in control antibody and from 4% to 20% in fremanezumab treated animals. This means, an increase in the number of animals would probably not increase the test strength essentially. Anyway, we are sorry not to be able to increase the number of experiments, because our experimental license does not cover this any more.
- How was it chosen that the CGRP measurement should be done on days 1, 3, and 10, while the immunohistochemistry was done on day 10 or 30 after fremanezumab treatment?
The maximal effect after fremanezumab regarding changes in CGRP release from the dura mater was seen after 10 days (Dux et al. 2022, ref. 11). Therefore we have primarily chosen this period for these experiments. Speculating that changes in expression pattern are even more delayed, we have added the 30 day period. At this time there was still a significant effect in lowering CGRP release after fremanezumab (Dux et al. 2022, ref. 11). This information has been added to the Discussion.
References:
In general, I recommend authors use more references to back their claims. I believe that adding more citations will help to provide better and more accurate background to this study.
We have added three more references to the Introduction and one additional to the Discussion conforming the statement of Paige et al. 2022: “…the available data indicate that rather functional than structural differences are underlying the greater sensitivity of female animals to CGRP and their higher susceptibility to inhibition of CGRP signalling, respectively [27]”.
Reviewer 3 Report
This is an animal model study analysing CGRP and components of its receptor in rats after administration of fremanezumab. The topic undertaken by authors is important considering only partially understood antiCGRP mAbs mechanism of action.
The study design and execution are very good, with appropriate methodical tools: control group, analysing for sex differences, addressing different CGRP-receptor components.
There are no major issues regarding this study.
Minor problems, that I have identified include:
- Abstract – fremanezumab is used in bot episodic and chronic migraine.
- Figure 3. – the ‘C” image could be larger and in the scale comparable to other images.
- Differences in numbers of specimens in particular comparisons are not comparhensively explained (e.g. why No. of specimens in males (Figure 4.) are different).
The results and conclusions of the study are valid and should be widely publicised to make public opinion and healthcare providers aware of a this new potential mechanism of action of anti ligand (CGRP) mAbs.
Author Response
Comments and Suggestions for Authors
This is an animal model study analysing CGRP and components of its receptor in rats after administration of fremanezumab. The topic undertaken by authors is important considering only partially understood antiCGRP mAbs mechanism of action.
The study design and execution are very good, with appropriate methodical tools: control group, analysing for sex differences, addressing different CGRP-receptor components.
There are no major issues regarding this study.
Minor problems, that I have identified include:
- Abstract – fremanezumab is used in bot episodic and chronic migraine.
We have added “and frequent episodic migraine” to the Abstract.
- Figure 3. – the ‘C” image could be larger and in the scale comparable to other images.
Figure 3C has the same scale as the other images, cf. size bars. There is nothing to see apart from the DAPI staining of nuclei and some faint background staining, which looks the same over the whole image. We prefer to show more of the CGRP immunostaining in A and B. However, for sake of clarity we have slightly increased 3C and for compensation slightly reduced the visible area of Fig. 3A and B. In addition, we have added stars to Fig. 3C marking (unvisible) neurons.
- Differences in numbers of specimens in particular comparisons are not comparhensively explained (e.g. why No. of specimens in males (Figure 4.) are different).
In our previous paper (Dux et al. 2022, ref. 11) the impact of fremanezumab in lowering CGRP release from the dura mater was higher in female animals. To ensure that we do not overlook a possible sex difference, in addition to the antibody difference, in CGRP immunoreactivity, we increased the number of female animals. However, a sex difference was not found here. This point has been mentioned in the Discussion.
Round 2
Reviewer 2 Report
Dear Authors,
I appreciate that the authors have taken my considerations into account, and all my concerns have been addressed. I accept the authors' answers.